# The influence of investor sentiment on the Chinese stock market amid COVID-19: An event study analysis

Yufei Sun[1]*, Chen Yang[2]

**1** School of Economics, Fuyang Normal University, Fuyang, Anhui, China, **2** East China Normal University, Putuo, Shanghai, China

* yufeisun666888@163.com

## Abstract

This study investigates the influence of investor sentiment on the Chinese stock market during the COVID-19 pandemic, using an event study analysis to examine data from December 2019 to December 2022. It aims to explore how investor sentiment, driven by news, social media, and economic uncertainties, has affected stock market performance during the pandemic. Data from 2005 to 2022 have been used to analyze abnormal and cumulative returns across key pandemic-related events, such as government interventions, lockdowns, and vaccine rollouts. The results show significant fluctuations in market returns driven by changes in sentiment. Positive sentiment, linked to government stimulus measures and vaccine announcements, led to positive market reactions, while negative sentiment, stemming from pandemic uncertainty, triggered market downturns. The study contributes to understanding the role of sentiment in market volatility, particularly in an emerging market like China, during periods of crisis. Accordingly, the study suggests multiple policy implications for policy makers.

## 1. Introduction

The COVID-19 outbreak in the Chinese stock market has caused large fluctuations linked to investor sentiments [1]. Some scholars have pointed out that investor sentiment has received quite some attention regarding change agents of markets. Still, its effect during a crisis such as the pandemic concerning the Chinese economy presents quite a challenge [2]. The pandemic has introduced new levels of risk and uncertainty while investors' confidence and market rationality have increased. These dynamics have destabilized the functionality of the Chinese stock exchange as investors' behaviors reacted to events and news rather than grounded technical factors and economic realities [3]. The monetary policy measures by the Chinese authorities, including monetary easing and administrative regulation, do not contribute to the solution of several emotional and psychological factors affecting share trading. One

**Data availability statement:** All relevant data are within the paper.

**Funding:** The author(s) received no specific funding for this work.

**Competing interests:** No.

of the critical problems has been the exacerbation of herd behavior during periods of increased uncertainty [4]. Notably, fear (or greed) has led investors to sell off their stocks in large volumes or, conversely, buy stocks in large volumes, leading to extreme market swings. Closely related to the lack of market stability and efficiency, this volatility is released in eradicating investors' dependence on the market to gain signals and insights into the underlying economy. The work of social media networks and other online platforms has only aggravated this factor, as the fast distribution of accurate or fake news affected investors' attitudes [5]. Negative information and false and banal narratives led to impulsive responses, although decisions were tactfully premeditated and more quickly than the regulatory bodies could impose measures.

Furthermore, the current COVID-19 pandemic also contributed to the increased disconnect between stock markets' performance and corporate values [6]. This has been the general public where actual economic conditions have had little effect on a company or stock's worth, certainly not as profound as the emotional swings of the market [7]. This has created a gap where long-term investors have been put off by volatility forces and short-term speculative sentiments, making it difficult to decide based on the actual market force [8]. Consequently, the intensity of the pricing system has been damaged, and it is no longer as efficient in allocating capital as it used to be in the Chinese stock market. Retail investors and institutional investors alike have also responded differently to the reality of the pandemic, which has only added to the issue of market segmentation [9]. The establishment investors, mainly using algorithm-based trading strategies, have benefited from impulse-driven oscillations. At the same time, the crowd, driven by fear and overheard optimism, has been badly beaten. The financialization effect also increased the gap between professional and non-professional investors, thereby leading to the deepening of economic inequalities and a reduction in market populism. Furthermore, some elements make it difficult to revive the Chinese stock market to the complete confidence of international investors [10]. Under China's 'incorrect' management of the virus outbreak and due to political conflicts between countries, foreign investors have sold their stakes, making the overall market less robust. This has had long-term implications for China's dream of turning the country into a financial center of the world. Accordingly, the study investigates the following research question;

*RQ: Does investor sentiment influence the Chinese stock market amid COVID-19?*

Our research contains multiple theoretical and practical contributions following the given research question. Theoretically, the present work has enriched the context of behavioral finance by conveying the dominance of investor sentiment in the dynamics of stock exchanges, especially during crisis periods, such as the COVID-19 outbreak. It has done so by adopting an event study method, which has enriched the understanding of the effect of investor sentiment on stock prices and the volatility of the Chinese market. The results have extended theories by illustrating that sentiment-based market action does not always conform to the prescribed financial and efficiency theory models. Further, this knowledge has enhanced the knowledge regarding the occurrence of market anomalies during crises and how external shocks

and sentiment affect markets. The integration of investor psychology and market behavior presented in this paper has provided a theoretical backdrop to similar occurrences in other emerging economies. Practically, the study has yielded policy implications for policymakers, regulators, and market players. It has therefore highlighted the necessity of timely and effective action to overcome irrational behavior of markets during crises if the identified mechanisms impact investor sentiment on the behavior of stock markets. Regulators have been urged to develop sentiment-sensitive policies wsuch as Some trading platforms have been brought forward to institutional investors for tweaking the trading algorithms to consider volatility resulting from sentiment. The retail investors have also had periodical education on financial literacy, thus enhancing structural stability and inclusion of investment in China. Fig 1 presents the structure of the study.

## 2. Review of literature

### 2.1 Investor sentiments, COVID-19 and Chinese stock market

There has been a significant development in the literature on the effects of investor sentiment on stock market movement, especially during the COVID-19 pandemic. Market sentiment is one of the most researched phenomena that explain several events in stock markets while contradicting classical theories. Behavioral finance analyses have shown that fear and over-optimism are fundamental feelings that impact investment decisions, influencing phenomena such as herd behavior and speculative bubbles. These effects have been more pronounced in these circumstances, with sentiment factors dominating price movements to the exclusion of fundamentals. The analysis of investor sentiment has been carried out by scholars who have noted that sentiment influences short-term stock price volatility that is unrelated to economic or corporate fundamentals. New external factors, like pandemics, have focused on how shifts in investor confidence are vulnerable and how much they affect financial markets. They have also been found to be intensifiers of sentiment in that they spread information, accurate and otherwise, that leads to trading in the market. This has put the regulatory authorities in a fix on how best to ensure that the market remains stable, especially during increased uncertainty.

Research on the Chinese stock market has highlighted that many retail investors are more sensitive to sentiment information. Speculative attitudes are also known to have been driven by market inefficiencies. The pandemic has been used to show some divergence between the stock market and the economy, where sentiment has become a top driver of prices. Following identifying the causes and consequences of sentiment-induced market fluctuations, the recent literature has extended to policy and regulation. Measures such as attempting to reduce the spread of false information, improve transparency and restore stability in investors' sentiments have been named necessary actions. In addition, analytics and sentiment-tracking tools have been recommended to monitor better and control market swings. In conclusion, the literature has been beneficial to readers in understanding the role of investor sentiment in the market and the fact that more theoretical development and intervention measures should be initiated to control the market fluctuation that it triggers.

During periods of extraordinary volatility in the financial markets, the likelihood of systemic disruption is much higher. [2] suggest that adverse events might trigger a series promoting negative expectations dissemination. Academic research has shown that economic and financial crises have impacted the stock market's stability. The global financial markets

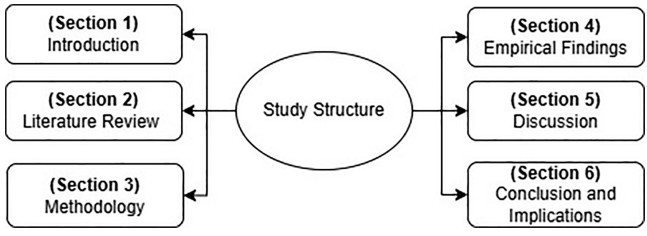

**Fig 1. Research structure.**

have declined due to the highly contagious COVID-19 virus. Although [11] investigated this phenomenon, our understanding of it remains limited. The urgency of further research in this area cannot be overstated. According to [12], the first documentation of COVID-19 emerged in late 2,019 in Wuhan, a city in the middle Hubei region of China. The official global public health emergency declaration was issued on January 30, 2020. As of early July 2020, there have been around 12,000,000 confirmed cases of COVID-19 and more than 500,000 deaths. By February 2020, the virus has spread to almost every continent. The rapid global spread of COVID-19 has led to widespread concerns about the economic prospects, resulting in several firms suspending their activities. The result was drastic fluctuations in value and sudden collapses in the stock market. According to [13], the stock markets in North America, Europe, and Asia saw their most significant single-day decline on record, followed by highly unpredictable daily price rebounds.

Due to the uncertain duration of the COVID-19 pandemic, accurately predicting its financial and economic impacts is challenging. This catastrophic calamity has intensified the unparalleled difficulties investors and portfolio managers encounter. However, it also presents an opportunity for innovative solutions. Companies have been compelled to use daring measures to embrace inventive investment methods to enhance profits while reducing risk due to the potential volatility of asset prices. The COVID-19 epidemic has shown the need to examine the substantial changes co-occurring in global markets meticulously. We must investigate this issue to fully understand the broader implications for the security of financial markets and its impact on other markets and to develop new strategies for managing such crises in the future [5,14,9,15]. Eminent financial analysts have dedicated significant effort and resources to thoroughly investigate the potential occurrence of contagion or spillover in alternative financial markets. Data from previous eras, especially those characterized by instability, are often juxtaposed with the findings of these inquiries. [1,16,17,18,19] Are prominent researchers in this particular domain? However, most of this research examines these subjects using a GARCH methodology to analyze the correlation and volatility within specific settings. Despite a significant systemic risk in global stock markets, its potential impact is often overlooked [1].

This study examines the transmission of COVID-19 as a significant systemic hazard, building upon the research conducted by [20]. Our primary emphasis is transferring risk outside the financial sector between international stock markets and the nations most severely affected by the pandemic. Given the current precarious condition of the stock market, we are exploring the potential for adverse consequences to have a broader reach. Gössling and colleagues [21] assert that the COVID-19 pandemic is a significant occurrence that induces extensive concerns. Several research conducted by [22,23,24,25] have shown that atypical occurrences may disrupt financial markets by altering individuals' perceptions of risk and reward. According to [26], the epidemic has caused people to become more cautious about taking risks and increased the availability of money. This has led to substantial fluctuations in high-risk investments. The COVID-19 pandemic has created a hazardous economic environment and significant health consequences due to worldwide lockdowns and adverse projections in several economic sectors [27]. According to [8], emotional elements are expected to have a significant impact throughout this pandemic period. According to financial sources, the epidemic has intensified market apprehension. Increased trading activity has led to substantial price fluctuations, as seen by (C. [28]). (Q. [28]) suggest that prudent investors often reallocate their funds from more volatile investments, such as equities, to more secure options like gold or government bonds. The interconnectedness of markets, which heightens susceptibility to extreme risks, and rapid divestment of assets all contribute to these results, highlighting the situation's complexity. There is a possibility of a broad spread of financial problems and unstable financial markets [29].

## 2.2 Highlighting research gap

The current literature analyzing investor sentiment's impact on stock markets, especially during crises such as the COVID-19 pandemic, has pointed out several essential research limitations. Although a vast number of papers attempt to examine the overall association of investors' sentiment and market fluctuation, most of such works fail to dedicate special attention to event-related dynamics in the Chinese market. Insufficient research focus has been given to how external

shocks like the current pandemic affected investor behavior in emerging economies, particularly China, where the markets are susceptible to sentiment influences, given the large number of retail investor participants. Moreover, prior studies have not researched how digital platforms and social media influence investor sentiment during crises. The availability of news and rumors has increased volatility, working in tandem with sentiment. Although China is an example of a country that has experienced such effects, few studies have examined the impact specifically in China.

Another gap is the poor incorporation of the sectoral impact on effects. Since the pandemic, industries have not been impacted equally; however, the effect of sentiment based on the Industry has received limited attention. Further, there is a dearth of research about using measures such as advance analytics and sentiment tracking to gauge and address sentiment-driven volatility in the Chinese stock market. The absence of a longitudinal study on sentiment effects beyond the crisis period has also been a gap, which has not clarified the substantive consequences of sentiment-induced fluctuation. Last, there is a theoretical integration deficiency between psychological and behavioral aspects of financial theories and financial models as identified in the existing studies. These gaps point to the importance of more extensive research that is more relevant to the country and more *trans*-disciplinary in its approach to better understand the nuances of investor sentiment in the context of China's stock market, especially in light of other extraordinary events such as the COVID-19 pandemic.

### 2.3 Theoretical framework

The present work has been based on the tenets of behavioral finance, which has revised the classical conception of the efficient market hypothesis, focusing on psychological factors affecting financial choices. H1 can be defined as investor sentiment – a key factor influencing stock markets, which has received extensive attention as a measure of market participants' overall bullish or bearish inclination. This work has incorporated behavioral finance theories, including herd behavior, overreaction, and the anchoring effect for analyzing sentiment-induced volatility. The framework established that during some of the crises, including the current COVID-19 pandemic, uncertainty and fear among investors tend to increase deviations from rational behavior, thus increasing market volatility. Furthermore, the research has used the efficient market hypothesis to compare market rationality and sentiment-related anomalies. While previous literature has presupposed that stock prices contain all available information, this paper has posited that sentiment has led to inefficiencies in the Chinese stock market mainly due to the high involvement of the retail investor. This study has also included the impact of digital platforms and social media on the work, suggesting that the fast dissemination of information during crises has accelerated the impulse trading theory. The framework developed in this paper has sought to integrate both behavioral and traditional financial theories to explain how investor sentiment works during external shocks. It has also recognized the need to fold sentiment metering devices and sentiment gauges into the market to provide empirical data on market trends. These theoretical assumptions have provided a strong framework for analyzing the specific features of the Chinese stock market during the the COVID-19 outbreak.

### 3. Data and methodology

#### 3.1 Data and sample selection criteria

The study has collected data from 2005 to 2022 on the Chinese stock market. For the event study examination, December 2019 to December 2022 is a COVID-19 outbreak. The A-share market, known for its stringent selection process, chooses companies to be included in the sample. The data is obtained through the most relevant sources of the Chinese market, including Wind Financial Terminal, Shanghai Stock Exchange reports, and the China Stock Market and Accounting Research (CSMAR) database. The sample selection for this study focuses on companies listed in the A-share market, known for its rigorous listing criteria, which ensures the inclusion of financially stable and reliable firms. The sample comprises publicly traded companies consistently listed from 2005 to 2022. For the event study, the analysis examines

the period from December 2019 to December 2022, covering the COVID-19 outbreak and its impact on the market. The selection criteria also include companies in the Shanghai Stock Exchange and those in the China Stock Market and Accounting Research (CSMAR) database. Companies with incomplete data or extreme market volatility unrelated to the pandemic were excluded to maintain data accuracy and reliability. Data sources include the Wind Financial Terminal and official stock exchange reports.

### 3.2 Variables and measures

The study contains topicality measures to analyze the effect of investor sentiment on the Chinese stock market during the COVID-19 crisis. In a theoretical sense, investor sentiment is defined as the overall state of people's minds in a market, which can influence market activity apart from fundamentals. This feeling usually translates into market anomalies, leading to changes that may not rationally relate to actualities. Since the testing of investor sentiment is done empirically, an Investor Sentiment Index (ISI) has been developed based on the sentiment analysis tools used on the financial news and social media. The equation of the index is as follows;

$$ISI_t = \frac{P - N}{P + N}$$

(1)

where $P$ represents positive words, and $N$ represents negative words in sentiment-rich documents, capturing the overall sentiment of investors during a specific period

$$AR_{i,t} = R_{i,t} - E\left(R_{i,t}\right)$$

(2)

To analyze stock market anomalies, the measures based on the excess returns (ER) have been used to indicate fluctuations of the actual stock returns compared to the situation on the regional stock markets, which is the expression of sentiment.

### 3.3 Event study: Planning and execution

The event study period is planned from December 2019 to December 2022. In the current research, the event study planning and implementation mechanism has been established carefully to comprehend the Chinese stock market during the COVID-19 situation from December 2019 to December 2022 and the role of investor sentiment. The events assumed to be central to the study include the first cases, primary policy on the epidemic, and coverage of vaccines, among others. These events have been classified into events to measure both the immediate and the delayed market responses. In the planning phase, the COVID-19 outbreak has been chosen to eliminate non-sentiment-related market impacts before and after each identified event. Information on market characteristics, including current price, traded volume, and volatility, has been obtained from Chinese databases and comprehensible sources such as CSMAR and WIND. It has used abnormal returns and cumulative Abnormal Returns (CAR) to compare the actual market performance to expected deviations in the COVID-19 outbreak during the implementation phase. In reflecting the investors' general emotional state, the quantitative sentiment measures have been extracted from the news articles, social media activity, and trading patterns. This rigorous mechanism has helped the study capture the complexity of how sentiment intersects with the market and provide sound conclusions about sentiment-induced anomalies in the Chinese stock market during the pandemic. The projected profits are computed using the Fama–French model. The ordinary least squares (OLS) regression is performed using the model (1)

$$R_{i,t} = \alpha + \gamma MKT_t + \delta SMB_t + \eta HML_t + \varepsilon_t$$

(3)

In this context, Ri, t refers to the rate of return of index I on date t during the specified estimation period. MKTt, SMBt, and HMLt represent the three components of the Fama-French model. The method for calculating abnormal returns is as follows:

$$AR_t = R_t - [\hat{\alpha} + \hat{\gamma}MKT_t + \hat{\delta}SMB_t + \hat{\eta}HML_t] \tag{4}$$

Where Rt represents the actual return on date t within the COVID-19 outbreak. To evaluate the total impact of an event within a designated time frame known as the "COVID-19 outbreak," we get the cumulative abnormal return (CAR) by adding up the individual abnormal returns using the formula.

$$CAR = \sum_{t=1}^{n} AR_t \tag{5}$$

Parametric tests are conducted to ascertain their significance after identifying and cumulative abnormal returns during the COVID-19 outbreak. Empirical data generally shows that the distributions of daily anomalous returns are related to a normal distribution, as [10] stated. According to the above facts, our null hypothesis asserts that the coefficient of the variable CAR is precisely zero. If the pandemic has a discernible positive impact on the stock price, the t-statistic should exhibit an enormous positive value, and vice versa. During the robustness test, we alter the length of the estimated period and the OLS regression model. The results remain consistent.

## 2.3 Panel regression model

Panel regression is superior to an event study for capturing the dynamic connection between dependent and independent variables across time. This is due to its ability to extract changes from panel data and mitigate estimation bias. Panel regression is used to reduce variability when measuring the influence of sentiment. The equation for the model can be expressed as:

$$R_{i,t} = \alpha_0 + \phi MKT_t + \delta SMB_t + \eta HML_t + \gamma_1 SENT_{i,t} + \gamma_2 EW_t + \gamma_3 L_i + \varepsilon_{i,t} \tag{6}$$

Whereas SENTi,t shows the sentiment index for the same stock, Ri,t at a specific point t reflects the stock's performance. A binary indication, the variable "EWt," indicates whether the COVID-19 outbreak exists or is absent. Should the event occur within the allocated period, its value is 0. Following the occurrence, the value is given the numerical value of one. The geographic location variable could assume 0, 2, and 1 values. Li explains the company's separate presence in Wuhan, other Hubei cities, and other parts of China. First, we do a regression analysis using SENT as the only independent variable. We then thoroughly include the long (L) and equal weight (EW) approaches to investigate the impacts of regions and reversals.

$$R_{i,t} = \phi_0 + \phi_1 SENT_{i,t} + \phi_2 SENT_{i,t} \times Feature_i + \gamma MKT_t + \delta SMB_t + \eta HML_t + \varepsilon_t \ln \tag{7}$$

Each group should utilize either fixed or random effects for hypothesis testing, according to the Hausman test. Whether the model should include random or fixed effects primarily depends on the p-value. The current work generated statistically significant p-values that indicated enough data to disprove the null hypothesis. We will so choose the fixed effects model. Furthermore, we improve the result's replicability and resilience using the appropriate generalized squares (FGLS) estimation technique.

## 4. Empirical findings

Table 1 presents concise statistical information for three periods: the estimate window, the COVID-19 outbreak, and the post-COVID-19 outbreak. The analysis focuses on three variables: market return, stock return, and sentiment. In Panel A,

the estimate window consists of 200 observations of the market return. The mean of these observations is 0.002, with a standard deviation (SD) of 0.008. The range of the observations spans from −0.029 to 0.022. The stock return has much higher volatility, with 424,829 observations, an average of 0.002, and a standard deviation of 0.022, ranging from −0.202 to 0.204. The sentiment, which has an exact number of observations as the stock return, has an average of 0.486 and a slightly higher standard deviation of 2.646. The values range from −29.880 to 22.464.

Panel B provides information on the COVID-19 outbreak, during which the market return is seen. In this case, the market return has been observed across 20 occurrences. The mean market return during this period is −0.006, with a standard deviation of 0.040. The most minor market return observed is −0.080, while the greatest is 0.020. Based on 42,486 data, the stock returns exhibit a mean of −0.006, a standard deviation of 0.046, and a range comparable to the estimate window. The sentiment exhibits a marginal decline in the average value to 0.442 and a marginal rise in the dispersion with a standard deviation of 2.828. Panel C displays the post-COVID-19 outbreak data, which indicates that the market return, calculated from 46 observations, has an average of −0.002 and a standard deviation of 0.028. The lowest and highest values recorded are −0.049 and 0.044, respectively. The stock returns, observed over 206,264 instances, have a mean value almost equal to zero and a standard deviation of 0.044. The sentiment score significantly rose to 0.628, with a standard deviation of 2.660, indicating a more positive perspective after the incident. Although exhibiting reduced volatility, the market returns transition from marginally positive to negative across periods, indicating the influence of events on market dynamics. The sentiment, although relatively more consistent than stock returns, has the highest level of fluctuation throughout the estimate and event periods, with a significant rise in the post-event period. This variance may suggest the shifting perspectives of market players and the subsequent adjustments made in the market after the events being considered. The summary statistics shown in Table 1 provide a fundamental comprehension of the patterns exhibited by market variables across various periods, paving the way for a more thorough examination of the factors that impact market returns, stock returns, and sentiment in the following sections and sparking your interest in the implications of these findings on market behavior.

Table 2 displays the cumulative abnormal return (CAR) for several event periods, analyzing the entire market, non-pharmaceutical sector, pharmaceutical Industry, and Hubei Province. The investigation encompasses two distinct time intervals: [0,9] and [20,46]. The average cumulative abnormal return (CAR) for the whole market for the [0,9] window is 0.008. The t-statistic for this CAR is 4.020, indicating statistical significance at the 1% level (shown by ***). This suggests a positive abnormal return over the beginning period. An abnormal return is a return on an investment that is higher or lower than the expected return, given the level of risk. However, during [20,46], the average CAR decreases to −0.026

**Table 1. Summary statistics.**

| Variable | Observations | Mean | SD | Min | Max |
|---|---|---|---|---|---|
| Panel A: Estimation Window | | | | | |
| Market return | 200 | 0.002 | 0.008 | −0.029 | 0.022 |
| Stock return | 424,829 | 0.002 | 0.022 | −0.202 | 0.204 |
| Sentiment | 424,829 | 0.486 | 2.646 | −29.880 | 22.464 |
| Panel B: Event Window | | | | | |
| Market return | 20 | −0.006 | 0.040 | −0.080 | 0.020 |
| Stock return | 42,486 | −0.006 | 0.046 | −0.204 | 0.204 |
| Sentiment | 42,486 | 0.442 | 2.828 | −20.922 | 22.024 |
| Panel C: Post-event Window | | | | | |
| Market return | 46 | −0.002 | 0.028 | −0.049 | 0.044 |
| Stock return | 206,264 | 0.000 | 0.044 | −0.209 | 0.204 |
| Sentiment | 206,264 | 0.628 | 2.660 | −20.644 | 22.628 |

**Table 2. Cumulative abnormal return for different COVID-19 outbreaks.**

| Event Windows | [0,9] | | [20,46] | |
|---|---|---|---|---|
| Indices | Average CAR | T-Stats. | Average CAR | T-Stats. |
| Overall Market | 0.008 | 4.020*** | −0.026 | −6.992*** |
| Non-Pharmaceutical Industry | 0.000 | −0.062 | −0.026 | −6.406*** |
| Pharmaceutical Industry | 0.220 | 26.482*** | −0.044 | −4.442*** |
| Shanghai Stock Market | 0.009 | 0.829 | −0.024 | −0.880 |

with a t-statistic of −6.992, which is similarly highly significant. This suggests a large negative anomalous return within the extended COVID-19 outbreak. In the non-pharmaceutical business, the average Compound Annual Return (CAR) is almost zero (0.000) during the period of [0,9], with a t-statistic of −0.062 that does not show statistical significance. Over the [20,46] frame, there is a persistent negative trend with an average Compound Annual Return (CAR) of −0.026 and a highly significant t-statistic of −6.406.

Fig 2 presents the panel cumulative tendencies of $ISI_t$ and $AR_{i,t}$, with the horizontal axis of the figure representing the time span between 2004 and 2023 and the vertical axis presenting the range of values, approximately between −0.06 and 0.12. The figure has two lines, which represent alternative indicators, since the blue line is for $AR_{i,t}$ and the orange line is for $ISI_t$. Typically, the two lines exhibit some degree of analogy of their direction, but with apparent deviations in some years, which serves as proof of their differences in dynamic development.

From the perspective of time series, during the period of 2004–2008, both curves hovered around the level of zero, which indicates that cumulative changes of both $ISI_t$ and $AR_{i,t}$ were quite limited and remained on a relatively stable level. Particularly between 2006 and 2008, the blue and orange lines hugged either side of the zero point closely, which meant that there was minimal divergence between the two indicators at this period. During the period from 2009 to 2012, orange line $ISI_t$ gradually increased and remained in the positive region, whereas blue line $AR_{i,t}$ dropped into the negative region, showing a significant negative accumulation in 2012. It indicates that $ISI_t$ maintained a consistently upward trend during this period, but $AR_{i,t}$ faced decreasing pressure, leading to a divergence period between the two trends.

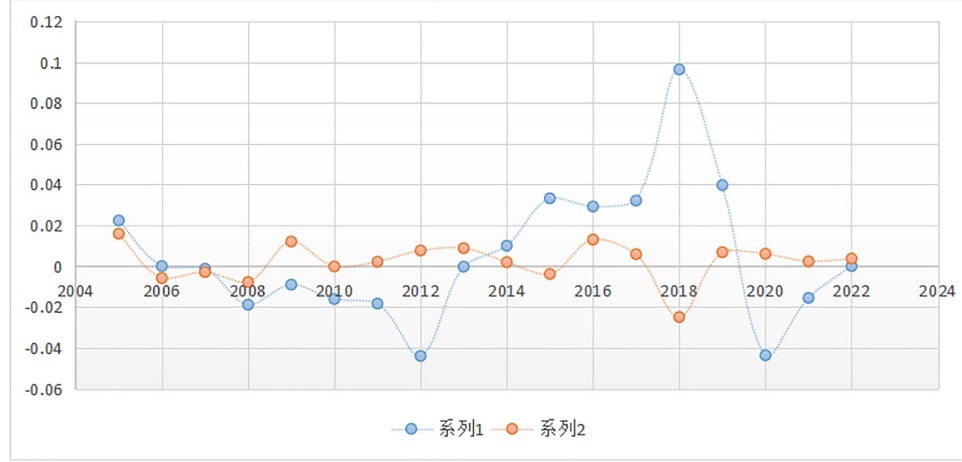

**Fig 2. Panel cumulative trends of $ISI_t$ and $AR_{i,t}$.** The orange line indicates $ISI_t$ the blue line indicates $AR_{i,t}$.

In the period of 2013–2016, the blue line slowly returned to the positive region, with some recovery of $AR_{i,t}$, while the orange line remained close to zero with relatively small fluctuations. The nature of the time is that the two indicators used to be close to each other, with neither of them exhibiting excessive volatility. However, starting from 2017, the blue line registered a sharp rise to the peak value of the entire time series in 2018 at levels above 0.1, which indicates an unwarranted accumulation of positive growth of $AR_{i,t}$ in 2018. On the other hand, the orange line suddenly fell below zero in 2018 to indicate a temporary decline. This sharp divergence suggests that the panel dynamics embodied by $AR_{i,t}$ and $ISI_t$ were considerably divergent at this stage.

The blue line subsequently declined sharply after 2019 and was close to a low of roughly −0.05 in 2020, before calmly reverting back to around zero, reflecting a typical pattern of over-volatility. In contrast, the orange line reflected increased stability, fluctuating by only small amounts around the zero level during 2019–2022, with considerably less variance compared to the blue line. This occurrence indicates that, in comparison to the volatile fluctuations of $AR_{i,t}$, the trend in cumulative $ISI_t$ is less significant, which means that it may have stronger mean-reversion properties or intrinsic stability.

This indicates sustained underperformance relative to the market over this time. In the [0,9] window, the pharmaceutical business has a distinct pattern, characterized by a substantial average CAR of 0.220 and a t-statistic of 26.482. These findings suggest strong positive abnormal returns. However, there is a reversal in this pattern within the [20,46] timeframe. The average CAR is −0.044, statistically significant, with a t-statistic of −4.442. This indicates a reduction in performance with time. Hubei Province has a relatively low average Compound Annual Growth Rate (CAR) of 0.009 during the period of [0,9]. The t-statistic of 0.829 indicates that this growth rate is not statistically significant. Within 20–46, the average CAR (cumulative abnormal return) decreases to −0.024. The t-statistic of −0.880 is not statistically significant, suggesting that the changes in abnormal returns are not as noticeable as the other indices. The data demonstrate clear and specific patterns of aberrant returns in several industries and areas during the event periods. The market first exhibits a favorable reaction but then experiences a downward trend, and the non-pharmaceutical sector continually performs below expectations.

Conversely, the pharmaceutical business has substantial advantages in the immediate period but encounters difficulties in the future. The events in question have reasonably consistently impacted Hubei Province, with slight changes observed. The research highlights the diverse influence of external events on various sectors and locations, offering valuable insights into the ever-changing nature of market reactions. Table 3 examines cumulative abnormal returns (CAR) across multiple sectors over specific event periods and the corresponding t-statistics and weights. With 298 observations, the pharmaceutical business has a notably high average Cumulative Abnormal Return (CAR) of 0.220 and a significant t-statistic of 26.482, which accounts for 24.22% of the total weight. This signifies a robust and favorable abnormal return, emphasizing the sector's ability to withstand and gain advantages from the analyzed occurrences. The production of computers, communication devices, and other electronic equipment, based on 294 data points, shows an average Compound

**Table 3. Analysis of accumulated anomalous return across various sectors at specific event periods.**

|  | Obs. | Average CAR | T-Stats. | Weight |
|---|---|---|---|---|
| *Pharmaceutical Industry* | 298 | 0.220 | 26.482*** | 24.22% |
| *Manufacture of computers, communication and other electronic equipment* | 294 | 0.044 | 4.889*** | 9.46% |
| *Software and information technology services* | 288 | 0.044 | 6.269*** | 8.62% |
| *Real estate* | 208 | −0.062 | −20.848*** | 6.49% |
| *Manufacture of special purpose machinery* | 288 | 0.026 | 2.849*** | 4.44% |
| *Manufacture of chemical raw materials and chemical products* | 204 | 0.026 | 2.098** | 2.98% |
| *Internet and related services* | 49 | 0.062 | 2.848*** | 2.44% |
| *Business Service Industry* | 44 | −0.068 | −6.696*** | 2.40% |
| *Other Industries* | 2,929 | −0.020 | −6.002*** | 42.26% |

Annual Return (CAR) of 0.044. The corresponding t-statistic of 4.889 is statistically significant, indicating a strong relationship. This sector contributes 9.46% to the overall weight. This Industry also encounters favorable abnormal returns due to heightened demand for technology and communication tools during the specified periods. The software and information technology services sector has a mean compound annual growth rate (CAR) of 0.044, with a t-statistic of 6.269. This analysis is based on 288 observations representing 8.62% of the total weight. This indicates a favorable reaction from the market, maybe due to increased dependence on digital services. The real estate sector has been observed 208 times and has an average CAR (cumulative abnormal return) of −0.062, indicating a negative trend. The t-statistic of −20.848 is highly significant, suggesting a strong relationship.

The real estate sector also accounts for 6.49% of the total weight. The underperformance indicates adverse effects on the real estate sector caused by market concerns or disruptions. The production of specialized equipment, based on 288 data, shows an average Compound Annual Growth Rate (CAR) of 0.026. The t-statistic of 2.849 is statistically significant, indicating a strong relationship. This sector contributes 4.44% to the overall weight. However, this signifies a favorable reaction with less prominent gains than other industries. The production of chemical raw materials and chemical products, based on 204 observations, has an average Compound Annual Return (CAR) of 0.026 and a statistically significant t-statistic of 2.098, corresponding to 2.98% of the total weight. The Industry is now seeing favorable financial gains, maybe due to a rise in the demand for chemical goods. The internet and associated services sector, with 49 observations, has an average Compound Annual Return (CAR) of 0.062 and a statistically significant t-statistic of 2.848, representing 2.44% of the total weight. This favorable outcome might be attributed to the substantial increase in internet activity. In the business service sector, 44 observations. The average CAR is −0.068, negative, and the t-statistic is −6.696, which is highly significant. This Industry contributes 2.40% to the weight. The sector's underperformance indicates the difficulties it is encountering. The other industries, consisting of 2,929 observations, exhibit an average CAR of −0.020, which is harmful, and a highly significant t-statistic of −6.002. These findings suggest that about 42.26% of the weight is attributed to a typically adverse reaction over various sectors. The findings demonstrate the diverse responses of different industries to events, with technology and medicines exhibiting resilience and favorable financial outcomes while real estate and business services encounter difficulties.

Table 4 displays the results of comparative tests conducted on industries and areas across three specific periods: [−120,-21], [0,9], and [10,45]. The investigation examines explicitly the pharmaceutical business and non-pharmaceutical industries, emphasizing differences in cumulative abnormal returns (CAR) and their degrees of statistical significance (P-values). Within the time frame of [−120,-21], the pharmaceutical business has a Compound Annual Growth Rate (CAR) of 0.024, which is not statistically significant according to traditional standards. Within the period from 0 to 9, the pharmaceutical business shows a notably negative CAR (cumulative abnormal return) of −0.042. The P-value suggests that this result is statistically significant at the 1% level. This indicates a negative response to the events that precede the significant event period.

However, within the same context, the Industry undergoes a favorable change with a Compound Annual Growth Rate (CAR) of 0.220, indicating a substantial enhancement in performance, backed by a very significant P-value. This suggests a strong and favorable reaction during the event, either because of industry-specific advancements or a higher demand for pharmaceutical items. Within 10–45, the pharmaceutical business's CAR (cumulative abnormal return) decreases to

**Table 4. Comparative analyses of regions and sectors.**

| Event Windows | [−120,-21] | | [0,9] | | [10,45] | |
|---|---|---|---|---|---|---|
| Indices | CAR | P-Value | CAR | P-Value | CAR | P-Value |
| Pharmaceutical industry | 0.024 | −0.042*** | 0.220 | −0.220*** | −0.026 | 0.028* |
| Non-Pharmaceutical | −0.008 | | −0.000 | | −0.044 | |

−0.026. This change is statistically significant at the 5% level, as shown by a P-value of 0.028. (*) This signifies a decline in performance after the incident, emphasizing the Industry's unpredictable reaction across several periods.

In contrast, the non-pharmaceutical sector did not exhibit noteworthy cumulative abnormal returns (CARs) during the examined periods. The CAR in the window [−120,-21] is −0.008, and there is no significant P-value connected with it. Similarly, within the time interval of [0,9], the cumulative abnormal return (CAR) is effectively zero, with a P-value that is not statistically significant. This indicates a neutral reaction during the early phase of the event. The CAR (Cumulative Abnormal Return) for non-pharmaceutical industries inside the [10,45] timeframe is −0.044. However, no P-value is supplied to assess the statistical significance. The negative CAR indicates a possible unfavorable effect during this timeframe, indicating a failure to recover or ongoing difficulties after the occurrence. The research highlights the pharmaceutical Industry's agile reaction to events, characterized by early adverse effects followed by robust positive gains and subsequent downturns. On the other hand, non-pharmaceutical industries show reasonably consistent, if less noticeable, reactions. These results emphasize the varied and diverse responses of markets in various sectors and periods. They provide valuable information on the distinctive dynamics of each Industry and the possible elements that affect performance throughout different parts of the COVID-19 outbreak. Table 5 presents the findings of a panel data regression analysis that investigates the impact of sentiment, reverse, and geography on market behavior. The analysis employs fixed effects (FE) and feasible generalized least squares (FGLS) approaches. The study uncovers noteworthy results about all three impacts. The sentiment impact has a consistently positive and highly significant correlation with market performance, as shown by SENTI coefficients of 0.006 and 0.004 for fixed effects (FE) and feasible generalized least squares (FGLS) models, respectively. The t-statistics, which are exceptionally high at 224.66 and 224.86, indicate a substantial statistical significance. The findings highlight the crucial importance of market mood in shaping stock returns, suggesting that positive sentiment is

**Table 5. Endogenity analysis.**

| Measures | 2SLS | | Replace the Variables | |
| --- | --- | --- | --- | --- |
| C | 0.114* | 0.327* | 0.646* | 0.021* |
| | (0.002) | (0.004) | (0.008) | (0.001) |
| SENTi | 0.515* | 0.802* | 0.139* | 0.585* |
| | (0.006) | (0.002) | (0.005) | (0.007) |
| MKT | 0.705* | 0.942* | 0.656* | 0.192* |
| | (0.006) | (0.004) | (0.006) | (0.005) |
| SMB | 0.387* | 0.925* | 0.661* | 0.723* |
| | (0.001) | (0.002) | (0.001) | (0.004) |
| HML | 0.617* | 0.561* | 0.128* | 0.911* |
| | (0.003) | (0.006) | (0.008) | (0.005) |
| EW | 0.451* | 0.591* | 0.438* | 0.024* |
| | (0.017) | (0.007) | (0.005) | (0.003) |
| $ISI_t$ | 0.209* | 0.429* | 0.201* | 0.795* |
| | (0.005) | (0.004) | (0.001) | (0.007) |
| $AR_{i,t}$ | 0.426* | 0.866* | 0.436* | 0.876* |
| | (0.000) | (0.001) | (0.003) | (0.009) |
| $R_{i,t}$ | 0.741* | 0.757* | 0.958* | 0.336* |
| | (0.007) | (0.002) | (0.003) | (0.005) |
| $E(R_{i,t})$ | 0.948* | 0.037* | 0.981* | 0.241* |
| | (0.001) | (0.008) | (0.005) | (0.008) |
| Wald Test | 1892.32 | 2451.11 | 1719.19 | 2204.14 |
| Significance | 0.001 | 0..009 | 0.000 | 0.003 |

associated with greater returns. The market's strong reaction to changes shows the reversal impact, as indicated by MKT coefficients of 2.026 and 2.028 for FE and FGLS, respectively.

Table 6 investigates the impact of investor sentiment upon returns over two phases: pre- and post-epidemic. This examination utilize fixed effects (FE) and feasible generalized least squares (FGLS) estimation approaches. $SENT_{i,t}$'s coefficient persistently takes on a statistically significant and positive number across all specifications of the model, confirming the presence of a strong and significant positive influence of investor sentiment upon returns. The pre-epidemic period ([−100, −1]) registers investors' coefficients at 0.004 and 0.002, respectively, for FE and FGLS. The post-epidemic period ([0, 45]), by contrast, registers a modest increment of the coefficients to 0.006 and 0.004, respectively, for FE and FGLS. The pattern suggests that the impact of investor sentiment upon equity returns has grown after the epidemic, signaling intense market sensitivity to behavioral and psychological variables both at the nexus of the crisis and across the post-crisis recovery process. The control variables perform expectedly: the market factor, MKT, always registers a positive and statistically significant outcome across all regression estimates, reinforcing that aggregate market movement explains variations of returns. Likewise, the size factor, SMB, always demonstrates positive significance, reflecting superior company performance among small-cap companies. The reverse, the value factor, HML, registers significantly negative correlation, indicating growth stocks beat values stocks, particularly across the epidemic period. The intercept term persistently retains significant negativity, reflecting minor downward return explained adjustment. The post-epidemic period registers growth of the model's explanation capability, with increased adjusted $R^2$ increment of 0.282 to 0.488, reflecting that the incorporation of sentiment and standard risk variables explains more of the return variation post-epidemic. In conclusion, Table 6 reflects that sentiment is a significant determinant of returns before and after the epidemic, with its influence growing after the epidemic occurrence.

Additionally, the t-statistics of 288.82 and 288.86 further corroborate the high importance of these coefficients. This suggests that market returns are responsive to overall market trends, underscoring the need to understand reversal dynamics. The size factor, SMB, has a strong and statistically significant positive association with market performance. The fixed effects (FE) and feasible generalized least squares (FGLS) coefficients are 0.626 and 0.628, respectively. The corresponding t-statistics are 66.40 and 66.66. This emphasizes the impact of small-cap equities on the overall market

**Table 6. Regression findings of panel data analysis comparing sentiment levels before and after the epidemic.**

| | [−100,-1] | [0,45] | [−100,-1] | [0,45] |
|---|---|---|---|---|
| | FE | FE | FGLS | FGLS |
| SENTi,t | 0.004*** | 0.006*** | 0.002*** | 0.004*** |
| | (−294) | (−224.66) | (−292.08) | (−224.86) |
| MKT | 2.020*** | 2.020*** | 2.024*** | 2.022*** |
| | (−299.69) | (−288.82) | (−299.82) | (−288.86) |
| SMB | 0.844*** | 0.604*** | 0.848*** | 0.608*** |
| | (−220.44) | (−64.62) | (−220.86) | (−64.86) |
| HML | −0.202*** | −0.208*** | −0.0996*** | −0.220*** |
| | (−22.94) | (−6.09) | (−22.68) | (−6.64) |
| α0 | −0.002*** | −0.002*** | −0.002*** | −0.002*** |
| | (−46.68) | (−26.44) | (−44.44) | (−24.89) |
| adj.R−sq | 0.282 | 0.488 | | |
| F | 40484.9 | 28404.2 | | |
| N | 428,902 | 84,466 | 428,902 | 84,466 |
| F | 2988.82 | 864.88 | | |
| P-value | 0.000*** | | 0.000*** | |

performance, indicating that smaller companies may have more significant returns throughout the analyzed periods. The value component, represented by HML, negatively correlates with market returns. The coefficients for FE and FGLS are −0.068 and −0.080, respectively, with corresponding t-statistics of −2.68 and −4.28, suggesting statistical significance (***). This indicates that value equities may perform worse than growth stocks during the analyzed event periods. The region's impact incorporates location-specific characteristics, primarily emphasizing Wuhan City and neighboring areas within Hubei.

Nevertheless, the coefficients about these locations lack statistical significance, indicating that geographical factors do not significantly influence market returns in this particular scenario. The constant term $\alpha_0$ exhibits a negative and significant value, measuring −0.002 for fixed effects (FE) and feasible generalized least squares (FGLS) models. The highly substantial t-statistics suggest a persistent negative return as a baseline, even when controlling for other variables. The adjusted R-squared value of 0.488 for both models suggests a considerable explanatory capacity, effectively capturing a significant proportion of the market returns' variability. These results emphasize the substantial impact of sentiment, market dynamics, and size on market performance, indicating that geographical characteristics may have a less critical function in this context. Table 7 displays the findings of a panel data regression analysis examining sentiment's impact before and after the pandemic. The study employs fixed effects (FE) and feasible generalized least squares (FGLS) methods. The investigation encompasses two time intervals: [−100,-1] and [0,45]. During the period before the pandemic (−100,-1), the sentiment coefficient was 0.004 using the fixed effects (FE) model and 0.002 using the feasible generalized least squares (FGLS) model. Both coefficients exhibit a high significance level, as shown by their respective t-statistics of −294 and −292.08. This indicates a robust and reliable correlation between emotion and market returns before the pandemic (***). The findings suggest that mood significantly and positively influenced returns over this period, emphasizing its crucial role in shaping market dynamics. Throughout the pandemic period [0,45], there is an observed rise in the sentiment

**Table 7. Regression findings of panel data analysis on sentiment, reversal, and regional effects.**

| | Sentiment Effect | | Reverse Effect | | Region Effect | |
|---|---|---|---|---|---|---|
| | FE | FGLS | FE | FGLS | FE | FGLS |
| SENTi | 0.006*** | 0.004*** | 0.006*** | 0.004*** | 0.006*** | 0.004*** |
| | (224.66) | (224.86) | (224.89) | (224.08) | (224.89) | (224.08) |
| MKT | 2.020*** | 2.022*** | 2.026*** | 2.028*** | 2.026*** | 2.028*** |
| | (288.82) | (288.86) | (288.62) | (288.68) | (288.62) | (288.68) |
| SMB | 0.604*** | 0.608*** | 0.626*** | 0.628*** | 0.626*** | 0.628*** |
| | (64.62) | (64.86) | (66.40) | (66.66) | (66.40) | (66.66) |
| HML | −0.208*** | −0.220*** | −0.068*** | −0.080*** | −0.068*** | −0.080*** |
| | (−6.09) | (−6.64) | (−2.68) | (−4.28) | (−2.68) | (−4.28) |
| EW | | | −0.002*** | −0.002*** | −0.002*** | −0.002*** |
| | | | (−9.28) | (−9.42) | (−9.28) | (−9.42) |
| Location | | | | | | |
| Other Hubei regions | | | | | 0.2431 | −0.000 |
| | | | | | (0.345) | (−0.29) |
| Wuhan City | | | | | 0.2461 | −0.002 |
| | | | | | (0.1242) | (−0.88) |
| $\alpha_0$ | −0.004*** | −0.002*** | −0.002*** | −0.002*** | −0.002*** | −0.002*** |
| | (−26.44) | (−24.89) | (−4.84) | (−4.92) | (−4.84) | (−4.86) |
| adj.R−sq | 0.488 | | 0.488 | | 0.488 | |
| F | 28404.2 | | 24966.8 | | 24966.8 | |
| N | 84,466 | 84,466 | 84,466 | 84,466 | 84,466 | 84,466 |

effect. The FE model shows a coefficient of 0.006, while the FGLS model shows a coefficient of 0.004. Both coefficients are statistically significant, with t-statistics of −224.66 and −224.86, respectively. This suggests that the influence of mood on market returns became stronger during the epidemic, indicating a heightened responsiveness to investor attitude in times of uncertainty.

Table 8 examines the role of sentiment by sector, and demonstrates sectoral heterogeneity in the contribution to returns. The baseline coefficients on SENT$_{i,t}$ remain positive and significant (0.005 with FE and 0.004 with FGLS), confirming that overall sentiment has a tendency to enhance returns across the equity market. But then adding interactions of sentiment with industry dummies, one can notice a pattern: most traditional and natural resource-based industries report negative and significant coefficients. Specifically, oil and gas extraction, petroleum and nuclear power refining, road transport, rail transport, and coal mining and washing all experience negative sentiment interactions, indicating that in all these sectors heightened sentiment lowers returns or at least reduces the positive effect achieved elsewhere. This means investor sentiment to punish those sectors associated with fossil fuels, high energy consumption, or pandemic-affected transport, as consistent with a structural trend for capital flowing out of "old economy" sectors. Other industries not listed here benefit directly from sentiment without the same negative effects, reinforcing the argument that sentiment favors growth-oriented and innovative sectors. The size factor (SMB) is again highly positive, supporting its presence as is the market factor (MKT), which is highly significant and positive, confirming its leadership status. The value factor (HML) is again negative, supporting the finding that growth companies were most positively impacted by sentiment forces. The constant is again small, negative,

**Table 8. Regression findings of panel data analysis for equities across several sectors.**

|  | FE | FGLS |
|---|---|---|
|  | $R_{i,t}$ | $R_{i,t}$ |
| SENT$_{i,t}$ | 0.005*** | 0.004*** |
|  | (8.09) | (7.96) |
| SENT$_{i,t}$ × Industry$_i$ |  |  |
| Oil and gas extraction | −0.004*** | −0.004*** |
|  | (−3.23) | (−3.25) |
| Petroleum and nuclear power processing | −0.004*** | −0.003** |
|  | (−2.76) | (−2.42) |
| Road transport | −0.004*** | −0.003*** |
|  | (−4.79) | (−4.42) |
| Railway transport industry | −0.004** | −0.003* |
|  | (−2.00) | (−1.84) |
| Coal mining and washing | −0.006*** | −0.003*** |
|  | (−4.43) | (−4.04) |
| MKT$_t$ | 1.014*** | 1.015*** |
|  | (184.07) | (184.16) |
| SMB$_t$ | 0.566*** | 0.571*** |
|  | (49.99) | (50.35) |
| HML$_t$ | −0.068*** | −0.080*** |
|  | (−3.11) | (−3.69) |
| $\varphi 0$ | −0.003*** | −0.003*** |
|  | (−25.22) | (−23.46) |
| adj. R−sq | 0.488 |  |
| F | 888.0 |  |
| N | 68,083 | 68,083 |

and significant, and the adjusted $R^2$ is solid at 0.488 for both FE and FGLS, which clearly indicates good explanatory power. Briefly, Table 8 shows that while sentiment has a net positive effect on returns, it is not uniformly distributed between sectors: growth and small-cap stocks are assisted more, while mature industry groups such as energy and transport are relatively penalized, underlining the selective nature of sentiment-driven capital allocation.

Across both eras, the market factor (MKT) consistently and significantly influences returns favorably. During the time before the pandemic, the MKT coefficients for FE and FGLS were 2.020 and 2.024, respectively. These coefficients were accompanied by very significant t-statistics of −299.69 and −299.82. During the pandemic, the MKT coefficient maintains significance, with values of 2.020 and 2.022 for fixed effects (FE) and feasible generalized least squares (FGLS). The t-statistics for these coefficients are −288.82 and −288.86. These data highlight the significant impact of market fluctuations on investment returns, which remain unaffected by the epidemic. Before the pandemic, the size factor (SMB) had a notable beneficial impact. The fixed effects (FE) coefficients and feasible generalized least squares (FGLS) were 0.844 and 0.848, respectively. The corresponding t-statistics were −220.44 and −220.86. Amidst the epidemic, the influence diminishes, with coefficients of 0.604 and 0.608 for fixed effects (FE) and feasible generalized least squares (FGLS), respectively, but remains statistically significant. This suggests that the size factor has a lower impact on returns during the pandemic. The value component (HML) negatively correlates with returns, as shown by the pre-pandemic coefficients of −0.202 for FE and −0.0996 for FGLS, accompanied by significant t-statistics of −22.94 and −22.68. There is a continued negative impact during the pandemic, as shown by the coefficients of −0.208 and −0.220 for fixed effects (FE) and feasible generalized least squares (FGLS), respectively. These coefficients are statistically significant, with t-statistics of −6.09 and −6.64. This indicates that value equities consistently perform worse than growth stocks in both periods. The constant component ($\alpha_0$) is consistently harmful and statistically significant, showing a baseline negative return. The coefficients of −0.002 are seen throughout both periods and models. The adjusted R-squared values of 0.282 and 0.488 for the pre-pandemic and pandemic eras indicate an enhanced ability to explain the data during the pandemic. This suggests that the market has become more responsive to the analyzed elements.

As shown in Table 9, all the results are highly significant, which proves our theory.

Table 10 presents the findings of a panel data regression analysis on stocks from various sectors. It investigates the impact of investor sentiment (SENT) on stock returns ($R_{i,t}$) using Fixed Effects (FE) and Feasible Generalized Least Squares (FGLS) models. The models demonstrate a statistically significant positive correlation between investor sentiment and stock returns. The coefficients for stock return are 0.005 and 0.004, with t-statistics of 8.09 and 7.96 for FE and FGLS, respectively. This indicates that higher investor sentiment is associated with greater stock returns. The interaction term SENT × Industry offers a detailed perspective on how sentiment impacts different industries. The coefficients for oil and gas extraction are −0.004, with t-statistics of −3.23 and −3.25, indicating a considerable negative influence of mood

**Table 9. Granger casualty diagnostics as a robustness analysis.**

| Null Hypothesis | Lag | F-value | Significance |
|---|---|---|---|
| $ISI_t$ not cause Granger causality | 1 | 0.174 | 0.001 |
| $AR_{i,t}$ not cause Granger causality | 2 | 0.937 | 0.005 |
| $R_{i,t}$ not cause Granger causality | 1 | 0.475 | 0.007 |
| $E(R_{i,t})$ not cause granger causality | 1 | 0.177 | 0.002 |
| $MKT_t$ not cause granger causality | 1 | 0.435 | 0.001 |
| $\delta SMB_t$ not cause granger causality | 1 | 0.272 | 0.003 |
| $\eta HML_t$ not cause granger causality | 1 | 0.885 | 0.000 |
| $CAR$ not cause granger causality | 1 | 0.946 | 0.001 |
| $AR_t$ not cause granger causality | 1 | 0.432 | 0.001 |

**Table 10. Regression findings of panel data analysis for equities with distinct attributes.**

| | FE | FGLS |
|---|---|---|
| | $R_{i,t}$ | $R_{i,t}$ |
| $SENT_{i,t}$ | 0.457 | 0.241 |
| | 0.004*** | 0.004*** |
| $SENT_{i,t} \times PE_i$ | 0.248 | 0.521 |
| | 0.001*** | 0.008*** |
| $SENT_{i,t} \times PB_{i,t}$ | 0.444 | 0.238 |
| | 0.007*** | 0.002*** |
| $SENT_{i,t} \times CMV_{i,t}$ | 0.570 | 0.621*** |
| | 0.005*** | 0.001*** |
| $SENT_{i,t} \times Year_i$ | 0.239 | 0.219 |
| | 0.004*** | 0.007*** |
| $SENT_{i,t} \times ISR_i$ | 0.634 | 0.403 |
| | 0.000*** | 0.008*** |
| $SENT_{i,t} \times NA_i$ | 0.455 | 0.645 |
| | 0.002*** | 0.564** |
| $MKT_t$ | 0.509 | 0.276 |
| | 0.000*** | 0.000*** |
| $SMB_t$ | 0.309 | 0.539 |
| | 0.002*** | 0.000** |
| $HML_t$ | 0.124 | 0.101 |
| | 0.041 | 0.680 |
| C | 0.471 | 0.512 |
| | 0.000*** | 0.003*** |
| adj. $R-sq$ | 0.488 | |
| F | 6024.2 | |
| N | 62842 | 62842 |

on returns in this Industry. Similar adverse sentiment effects are reported in the petroleum and nuclear power processing, road transport, and railway transport sectors, with coefficients hanging around −0.004 and varied degrees of statistical significance. These results indicate that industries strongly affected by external variables such as regulations and commodity prices may negatively respond to changes in investor mood. Within the coal mining and washing business, the interaction term has a more significant detrimental effect on returns, as shown by coefficients of −0.006 and −0.003. This suggests a more negative response to changes in sentiment. The market factor (MKT) strongly impacts stock returns across several sectors, with coefficients of 1.014 and 1.015, highlighting the significant significance of market fluctuations in determining returns. The size factor (SMB) positively correlates with stock returns, while the value factor (HML) indicates a negative impact. This implies that companies with large book-to-market ratios tend to perform poorly when considering sentiment when calculating returns. The intercept ($\phi_0$) is statistically significant and has a negative value of −0.003 for both models. This indicates a baseline negative return in the absence of any other factors. The adjusted R-squared value of 0.488 suggests that the model can account for almost 49% of the variability in stock returns. The F-statistic of 888.0 confirms that the regression is statistically significant. The substantial sample size (N) of 68,083 observations further reinforces the dependability and resilience of the findings, offering a complete perspective on the diverse influence of sentiment across different sectors.

## 5. Discussion on findings

The result of the present analysis sheds light on the critical role of investor psychology in the context of stock returns in China, particularly in light of the COVID-19 outbreak. Through the method of event study conducted for this research and measurement of ARs and CARs, the roles of investor sentiments in generating market fluctuations during several events of the pandemic are explained. The sentiment analysis of financial news, social media posts, and market data has provided more profound study results about sentiment, stock prices, and market trends. These findings show the varying investor sentiment during some crucial moments of the pandemic, including the virus outbreak in December 2019, the various governmental measures, the lockdowns, and the initial COVID vaccine roll outs. Similarly, investor sentiment seemed to replicate these events with having recorded high positive sentiments during government's release of stimulus measures and key points concerning the rollout of vaccines. At these times, the market showed positive abnormal returns, depicting the recovery and market stability from an investors perspective. Conversely, periods of uncertainty, especially during the early stage of the pandemic and periods of rising and falling infection rates, indicated negative investor sentiments, giving adverse stock market abnormal returns.

A significant conclusion that the authors drew from the analysis was the scale of sentiment-based activity concerning the more conventional mechanisms of the stock market's functioning. Under increased uncertainty, the Chinese stock market is characterized by higher volatility in response to sentiment factors than expected under the standard financial theory. Such excessive movements indicate that fear and optimism-driven sentiments were instrumental in price changes, hence inefficiency and overreaction to pandemic events. This finding supports the behavioral finance theory that investors undertaking actual activities do not act rationally based on the information gathered but instead are bounded by emotions. The CAR analysis computed on specific COVID-19 outbreaks also influenced the relative and, from the perspective of sentiment aroused, persistent length of market reactions. In particular, the findings suggest that market reactions were extended and lasted for multiple weeks after significant events occurred. For instance, soon after thGovernment put measures for locking down and banning travel, there was some negative change in the performance of the stock market, with negative sentiments generally carrying forward for several days. But when there were positive sentiments from vaccine rollout announcements, recovering was seen, though the scale and pace differed across industries. This appears to imply that even through the market moves in response to sentiment in the short-run, sentiment may not always define the long-run market stability. This late reaction of the stock market to some positivity like the vaccine distribution also makes it clear how sentiment is intertwined with the market. Such empirical evidence implies that although the expectations of investors play a major role in temporary shifts of markets, their longer-term impact is highly dependent on the changing circumstances of the economy.

There was also a difference observed in positive and negative influences, in different types of sectors. This study discovered that some industries offered better positive abnormal returns acting in response to Pollyanna shocks, meaning those related to technology and healthcare sales displayed greater positive sentiment regarding recovery and response to the pandemic. On the other hand, negative abnormal returns were, however, higher in sectors such as tourism, hospitality, and retail since the adverse news factors that impacted investors' sentiment in the COVID-19 period included matters like continued lockdowns, restrictions on movements, behavior changes, among others. This variation for the sector level stresses the necessity to consider the specifics of the economy when considering the sentiment at play, as the same changes of sentiment, impacted different sectors differently. Hybrid caveat: Thus, the investor sentiment in China has not been diverse across sectors, but has instead captured the different loyalties of various industries during the pandemic period. The indices of sentiment measurement based on data from financial news presentations, social networks, and trading activity also agree with the idea about the gravity of information dissemination for market sentiment. The analyse of sentiment on social media showed that many new shareholders in China, who are usually trading through pure-Internet brokers, overestimated the economic consequences of the pandemic, relying on the information they received from media

and social networks. This observation reveals the role of platforms and social media in influencing investors' perceptions in current-day markets. The fast dissemination of relevant and irrelevant information on these forums may result in market overreaction, strengthening the influence of sentiment in the variation of stock prices.

Furthermore, the results of this paper add to the accumulating literature on the impact of investor sentiment within the EM context, especially during crises or pandemics such as COVID-19. The analysis presented in the paper demonstrates that psychological and emotional factors play an essential role in forming markets. Basic models involving only facts and financial ratios cannot explain their activity during crises. Additionally, the results indicate that flow is sensitive to information and sentiment and that China-based retail investors are highly sensitive to both. The present study also has policy and regulatory implications for policy makers and financial regulators. Knowing the key factors that give rise to sentiment's effect on volatility helps formulate better and more efficient methods of correcting abuse in volatile markets and managing the general public's sentiment during crises. To that end, ignoring sentiment and not understanding that the walls of China are indeed baiting for investors is a misconception that, if addressed, will permit regulators to temper some form of investor hysteria. Given the above, this study has successfully established the nature and dynamics of sentiment and stock market behavior in China, especially amid the COVID-19 shock, and called for more resent studies on the influence of sentiment in emerging markets during a crisis period.

## 6. Conclusion, policy implications, and future research

### 6.1 Conclusion

In this paper, the author has looked at the impact of investor sentiment on the Chinese stock market during the COVID-19 crisis using event study analysis. It has shown that the cross-section of investor sentiment due to COVID-19 positive and negative socio-historical events affecting the world has affected the abnormalities in the stock return. Interventions from the government and announcements regarding vaccines boosted market performance; terminologies and changes in the market led to adverse performance during the pandemic. The study also underlines the significance of the sentiment factor in establishing market fluctuation that may deviate from the prevailing economic conditions. By acquiring new knowledge about investors' behavior during the global crisis and the nature of stock market dynamics during such an event, this study provides methodological and practical recommendations for improving investor and stakeholder behavior and minimizing their losses when managing stock market volatility during similar future events concerning the role of sentiment analysis.

### 6.2 Policy implications

In light of the conclusions made in this paper, several policy implications would be significant to governing China's stock market, especially during a crisis. The increasing impact of investor sentiment on the stock market during the COVID-19 pandemic highlights the need for better research on the role of sentiment in the stock market and improvement of measures towards regulating the market and protecting investors during crises. First, we show that government communication strategies must be appropriately constructed so the volatility related to investor sentiment can be controlled. This study has ascertained that the Chinese stock market responded highly sensitive to new government measures such as lockdowns and stimulus packages. To dampen such fluctuations and provide clear investment signals, the government should adopt plain language of communication with investors and make it timely. This would help decrease, which is a significant determinant of the other er-proliferation of the th. Since investors receive timely and consistent status on public health, wielding economic instruments, and possible future policy shifts, the government can help regulate speculative market motives. Second, it will be in the interest of the Chinese financial regulators to specifically increase financial innovation among retail investors because sentiments primarily influence them. It also calls for educational approaches for investors regarding information about the nature of markets and behavioral patterns, the establishment of risk controls,

and the impact of sentimentality. This can lessen the effect of substantially erratic market reactions propagated by market sentiment shifts ahead of comprehensive data. Similarly, relying on the results obtained, the study focuses on the need for information control in the information society. Since social media and news influenced investors' perceptions during the COVID-19 pandemic, governments should act to control the spread of financial news. It also noted that attempts to guide or alert investors to credible, fact-based news can help diminish overly emotional market swings. Further, measures that can be attributed to preventing volatility, like trading halts or circuit breakers, can also be used. If trading were paused when panic of herd behavior and emotions starts to rise, then regulators would be able to avert the situations that took place with a particular shout, which implies that the markets would be saved. They could also help restore confidence and guarantee market stability in the case of its emergence. Lastly, China's policymakers should persistently search for market reforms that provide an appropriate blend of retail and institution. Institutional investors use more rather fundamentals, so by taking their share, they can compensate for the emotions of other participants. The roles of institutional investors should be enhanced as this idea could lead to increased efficiency and stability in the market.

### 6.3 Limitations and future research

Despite the contribution of this research in understanding the impact of investor sentiment on Chinese stock during the COVID-19 pandemic, there are limitations that need to be acknowledged. Members of this research have exclusively targeted the Chinese market, meaning that the findings cannot be generalized to other emergent economies with different markets. Furthermore, the study mainly focuses on the private-sector sentiment data from social media and news, which may be insufficient to capture institutional investor behavior and the economic sentiment that affects sentiment. Subsequent studies may take other developing countries to the analysis and determine how sentiment affects stock markets in such global regions. Furthermore, extending the analysis to studying the effect of sentiment on market behavior in the post-crisis period and including sentiment indicators originating from various sources, including polls among investors and financial statements, may provide a more comprehensive picture of the investor's choice and stock market movements.

### Author contributions

**Investigation:** Yufei Sun.

**Methodology:** Yufei Sun.

**Project administration:** Yufei Sun, Chen Yang.

**Resources:** Yufei Sun, Chen Yang.

**Software:** Chen Yang.

**Supervision:** Chen Yang.

**Writing – original draft:** Yufei Sun.

**Writing – review & editing:** Yufei Sun.

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
