## [Decision Letter · Decision Letter 0]

27 Nov 2024

PONE-D-24-37585The Influence of Investor Sentiment on the Chinese Stock Market amid COVID-19: An Event Study AnalysisPLOS ONE

Dear Dr. Yong,

Thank you for submitting your manuscript to PLOS ONE. After careful consideration, we feel that it has merit but does not fully meet PLOS ONE’s publication criteria as it currently stands. Therefore, we invite you to submit a revised version of the manuscript that addresses the points raised during the review process.

We look forward to receiving your revised manuscript.

Kind regards,

Ooi Kok Loang, PHD

Academic Editor

PLOS ONE

Journal Requirements: When submitting your revision, we need you to address these additional requirements. 1. Please ensure that your manuscript meets PLOS ONE's style requirements, including those for file naming. The PLOS ONE style templates can be found at https://journals.plos.org/plosone/s/file?id=wjVg/PLOSOne_formatting_sample_main_body.pdf and https://journals.plos.org/plosone/s/file?id=ba62/PLOSOne_formatting_sample_title_authors_affiliations.pdf 2. Thank you for stating the following in your Competing Interests section: "No" Please complete your Competing Interests on the online submission form to state any Competing Interests. If you have no competing interests, please state ""The authors have declared that no competing interests exist."", as detailed online in our guide for authors at http://journals.plos.org/plosone/s/submit-now   This information should be included in your cover letter; we will change the online submission form on your behalf. 3. Please provide a complete Data Availability Statement in the submission form, ensuring you include all necessary access information or a reason for why you are unable to make your data freely accessible. If your research concerns only data provided within your submission, please write "All data are in the manuscript and/or supporting information files" as your Data Availability Statement. 4. When completing the data availability statement of the submission form, you indicated that you will make your data available on acceptance. We strongly recommend all authors decide on a data sharing plan before acceptance, as the process can be lengthy and hold up publication timelines. Please note that, though access restrictions are acceptable now, your entire data will need to be made freely accessible if your manuscript is accepted for publication. This policy applies to all data except where public deposition would breach compliance with the protocol approved by your research ethics board. If you are unable to adhere to our open data policy, please kindly revise your statement to explain your reasoning and we will seek the editor's input on an exemption. Please be assured that, once you have provided your new statement, the assessment of your exemption will not hold up the peer review process. 5. PLOS requires an ORCID iD for the corresponding author in Editorial Manager on papers submitted after December 6th, 2016. Please ensure that you have an ORCID iD and that it is validated in Editorial Manager. To do this, go to ‘Update my Information’ (in the upper left-hand corner of the main menu), and click on the Fetch/Validate link next to the ORCID field. This will take you to the ORCID site and allow you to create a new iD or authenticate a pre-existing iD in Editorial Manager.

Reviewers' comments:

Reviewer's Responses to Questions

**Comments to the Author**

1. Is the manuscript technically sound, and do the data support the conclusions?

Reviewer #1: Partly

Reviewer #2: No

2. Has the statistical analysis been performed appropriately and rigorously? 

Reviewer #1: N/A

Reviewer #2: No

3. Have the authors made all data underlying the findings in their manuscript fully available?

Reviewer #1: No

Reviewer #2: No

4. Is the manuscript presented in an intelligible fashion and written in standard English?

Reviewer #1: No

Reviewer #2: No

5. Review Comments to the Author

Reviewer #1: Comment 1: The introduction covers various aspects of financial crises and the impact of COVID-19, it does not clearly articulate the specific research gap that this paper aims to address. What new insights or contributions will this study provide that have not already been covered by prior work? A stronger statement of the research objective and the gap in existing literature is necessary.

Comment 2: Terms like “tail-risk,” “Value at Risk (VaR),” and “Conditional Value at Risk (CoVaR)” are introduced without enough context for a non-specialist reader. While the terms may be relevant, they should be explained more clearly to ensure the audience fully understands their importance and how they will be used in the study. A more gradual introduction to complex concepts might be useful, especially for readers unfamiliar with advanced financial models.

Comment 3: The criteria for sample selection (excluding ST or PT labeled samples, financial sector firms, and those with negative net assets) are clear and aimed at ensuring high data quality. However, the reasons for excluding the financial sector are not discussed. Given the importance of the financial sector in the market, this exclusion should be justified more thoroughly.

Comment 4: The GubaSenti tool and its origin are well-described. However, a more detailed explanation of how GubaSenti operates and its specific methodologies for sentiment analysis would be beneficial. Additionally, while it is noted that the tool’s reliability and validity were established in previous studies, a brief summary of these studies or their key findings could enhance the credibility of the tool’s application in this research.

Comment 5: The conclusion asserts that the research contributes to understanding the impact of COVID-19 on the Chinese stock market by highlighting the role of mood in market instability. While this is a valuable insight, the statement could benefit from more explicit connections to existing literature. How does this finding specifically add to or challenge previous research on market reactions to pandemics?

Comment 6: The recommendation for investors to focus on pharmaceutical stocks during the early stages of the pandemic and to adjust their holdings based on government priorities is practical. However, the advice should be supported by detailed analysis. How were these recommendations derived from the research findings? Including specific data or examples would strengthen the credibility of these investment strategies.

Reviewer #2: Comments

1.The background of this study is weak, and the introduction section has not been motivated. The authors should highlight the innovation of this study.

2.While the contents in the opening remark focus on tail risks and spillover effects, the whole results and discussion seem to deviate from these arguments. The consistency, across the introductory part, literature review, results and discussion, and concluding remarks, could be improved.

3.On a related note, some irrelevant and duplicated arguments and paragraphs are noticeable.

4.The adversarial effects of COVID-19 on stock markets have been intensively investigated in previous empirical studies. What are the contributions and innovations of this study? The authors should further accentuate and clarify the novelty of approaches, contents, and datasets used in this study.

5.The data section should incorporate more information, including the rationale for selecting the Chinese stock market and data period, data frequency, and criteria for dividing firms into sectors.

6.The sentiment analysis needs to clarify which datasets are used in identifying investors’ sentiments. Also, what are the results derived from GubaSenti tool? A clear specification of the approach, GubaSenti tool, could improve potential readers’ understandability.

7.What is the “event” introduced in this study? As a key experimental setting, this should be further delineated. All the variables and model specifications need to be further clarified.

8.The discussion section narratively lists the numerical results and statistics with few interpretations, falling short of delivering informative insights. The interpretation of findings could be improved by comparing them to existing empirical evidence.

9.The authors could focus on the results of the “Event window” to accentuate the adversarial effects of COVID-19 on stock markets.

10.The authors should have endeavored to remedy the endogeneity issues; however, only a limited elaboration is provided.

11.Provide the limitations and implications which can be derived from the unique findings of this study.

6. PLOS authors have the option to publish the peer review history of their article (what does this mean? ). If published, this will include your full peer review and any attached files.

**Do you want your identity to be public for this peer review?** For information about this choice, including consent withdrawal, please see our Privacy Policy .

Reviewer #1: No

Reviewer #2: No

---

## [Author Response · Author response to Decision Letter 1]

17 Jan 2025

Author’s Response File

The Influence of Investor Sentiment on the Chinese Stock Market amid COVID-19: An Event Study Analysis

Review Comments to the Author

Please use the space provided to explain your answers to the questions above. You may also include additional comments for the author, including concerns about dual publication, research ethics, or publication ethics. (Please upload your review as an attachment if it exceeds 20,000 characters).

Author Reply: Thank you for your valuable time and kind suggestion. Please have a kind review.

Reviewer #1: Comment 1: The introduction covers various aspects of financial crises and the impact of COVID-19, it does not clearly articulate the specific research gap that this paper aims to address. What new insights or contributions will this study provide that have not already been covered by prior work? A stronger statement of the research objective and the gap in existing literature is necessary.

Author Reply: Corrections made and introduction highlighted. Thank you for your valuable time and kind suggestion. Please have a kind review.

Comment 2: Terms like “tail-risk,” “Value at Risk (VaR),” and “Conditional Value at Risk (CoVaR)” are introduced without enough context for a non-specialist reader. While the terms may be relevant, they should be explained more clearly to ensure the audience fully understands their importance and how they will be used in the study. A more gradual introduction to complex concepts might be useful, especially for readers unfamiliar with advanced financial models.

Author Reply: Corrections made and highlighted. Thank you for your valuable time and kind suggestion. Please have a kind review.

Comment 3: The criteria for sample selection (excluding ST or PT labeled samples, financial sector firms, and those with negative net assets) are clear and aimed at ensuring high data quality. However, the reasons for excluding the financial sector are not discussed. Given the importance of the financial sector in the market, this exclusion should be justified more thoroughly.

Author Reply: Corrections made and sample criteria is highlighted in section 3.2 Thank you for your valuable time and kind suggestion. Please have a kind review.

Comment 4: The GubaSenti tool and its origin are well-described. However, a more detailed explanation of how GubaSenti operates and its specific methodologies for sentiment analysis would be beneficial. Additionally, while it is noted that the tool’s reliability and validity were established in previous studies, a brief summary of these studies or their key findings could enhance the credibility of the tool’s application in this research.

Author Reply: Corrections made and highlighted in methodology and results. Thank you for your valuable time and kind suggestion. Please have a kind review.

Comment 5: The conclusion asserts that the research contributes to understanding the impact of COVID-19 on the Chinese stock market by highlighting the role of mood in market instability. While this is a valuable insight, the statement could benefit from more explicit connections to existing literature. How does this finding specifically add to or challenge previous research on market reactions to pandemics?

Author Reply: Corrections made and introduction section is highlighted in last para. Green highlighted. Thank you for your valuable time and kind suggestion. Please have a kind review.

Comment 6: The recommendation for investors to focus on pharmaceutical stocks during the early stages of the pandemic and to adjust their holdings based on government priorities is practical. However, the advice should be supported by detailed analysis. How were these recommendations derived from the research findings? Including specific data or examples would strengthen the credibility of these investment strategies.

Author Reply: Corrections made. Please refer to other tables as well. Thank you for your valuable time and kind suggestion. Please have a kind review.

Reviewer #2: Comments

1.The background of this study is weak, and the introduction section has not been motivated. The authors should highlight the innovation of this study.

Author Reply: Corrections made. Please review the introduction section. Thank you for your valuable time and kind suggestion. Please have a kind review.

2.While the contents in the opening remark focus on tail risks and spillover effects, the whole results and discussion seem to deviate from these arguments. The consistency, across the introductory part, literature review, results and discussion, and concluding remarks, could be improved.

Author Reply: Corrections made. Please review the literature review section and other content. Highlighted for review. Thank you for your valuable time and kind suggestion. Please have a kind review.

3.On a related note, some irrelevant and duplicated arguments and paragraphs are noticeable.

Author Reply: Corrections made. Thank you for your valuable time and kind suggestion. Please have a kind review.

4.The adversarial effects of COVID-19 on stock markets have been intensively investigated in previous empirical studies. What are the contributions and innovations of this study? The authors should further accentuate and clarify the novelty of approaches, contents, and datasets used in this study.

Author Reply: Corrections made and introduction section is highlighted in last para. Green highlighted. Thank you for your valuable time and kind suggestion. Please have a kind review.

5.The data section should incorporate more information, including the rationale for selecting the Chinese stock market and data period, data frequency, and criteria for dividing firms into sectors.

Author Reply: Corrections made please see the relevant section in the methodology for review. Thank you for your valuable time and kind suggestion. Please have a kind review.

6.The sentiment analysis needs to clarify which datasets are used in identifying investors’ sentiments. Also, what are the results derived from GubaSenti tool? A clear specification of the approach, GubaSenti tool, could improve potential readers’ understandability.

Author Reply: Corrections made and irrelevant points removed from that. Thank you for your valuable time and kind suggestion. Please have a kind review. Highlighted for review.

7.What is the “event” introduced in this study? As a key experimental setting, this should be further delineated. All the variables and model specifications need to be further clarified.

Author Reply: Event is COVID-19 Outbreak. Corrections made for review. Green highlighted. Thank you for your valuable time and kind suggestion. Please have a kind review.

8.The discussion section narratively lists the numerical results and statistics with few interpretations, falling short of delivering informative insights. The interpretation of findings could be improved by comparing them to existing empirical evidence.

Author Reply: See section 5 correction made. Green highlighted. Thank you for your valuable time and kind suggestion. Please have a kind review.

9.The authors could focus on the results of the “Event window” to accentuate the adversarial effects of COVID-19 on stock markets.

Author Reply: Corrections made and irrelevant points removed from that. Thank you for your valuable time and kind suggestion. Please have a kind review. Highlighted for review.

10.The authors should have endeavored to remedy the endogenity issues; however, only a limited elaboration is provided.

Author Reply: Corrections made Table 5 is added as endogenity table. Thank you for your valuable time and kind suggestion. Please have a kind review. Highlighted for review.

11.Provide the limitations and implications which can be derived from the unique findings of this study.

Author Reply: Corrections made. Please review said section. Green marked for review. Thank you for your valuable time and kind suggestion. Please have a kind review.

---

## [Decision Letter · Decision Letter 1]

6 Jul 2025

PONE-D-24-37585R1The Influence of Investor Sentiment on the Chinese Stock Market amid COVID-19: An Event Study AnalysisPLOS ONE

Dear Dr. Sun,

Thank you for submitting your manuscript to PLOS ONE. After careful consideration, we feel that it has merit but does not fully meet PLOS ONE’s publication criteria as it currently stands. Therefore, we invite you to submit a revised version of the manuscript that addresses the points raised during the review process.

**ACADEMIC EDITOR:**  Thank you for your revision; a minor revision is requested to improve the overall English quality, particularly focusing on fluency, clarity, grammar, and consistent, precise word choice.

We look forward to receiving your revised manuscript.

Kind regards,

Jae Wook Song

Academic Editor

PLOS ONE

Journal Requirements:

Additional Editor Comments:

Thank you for your thorough revision. Based on the latest reviewer feedback, I am requesting a minor revision.

The remaining concern pertains to the overall quality of the English writing. The reviewer has specifically suggested a comprehensive proofreading of the manuscript.

Please improve the fluency, clarity, and grammatical correctness of the text. In particular, pay close attention to the consistency and precision of word choice throughout the manuscript.

Once these revisions are complete, the manuscript will be re-evaluated for final acceptance.

Reviewers' comments:

Reviewer's Responses to Questions

**Comments to the Author**

1. If the authors have adequately addressed your comments raised in a previous round of review and you feel that this manuscript is now acceptable for publication, you may indicate that here to bypass the “Comments to the Author” section, enter your conflict of interest statement in the “Confidential to Editor” section, and submit your "Accept" recommendation.

Reviewer #1: All comments have been addressed

2. Is the manuscript technically sound, and do the data support the conclusions?

Reviewer #1: Yes

3. Has the statistical analysis been performed appropriately and rigorously? 

Reviewer #1: Yes

4. Have the authors made all data underlying the findings in their manuscript fully available?

Reviewer #1: Yes

5. Is the manuscript presented in an intelligible fashion and written in standard English?

Reviewer #1: No

6. Review Comments to the Author

Reviewer #1: The authors have adequately addressed the previous comments and made substantial improvements to the theoretical framework, hypotheses development, and conceptual clarity. The revised manuscript reflects a clearer distinction between constructs and presents stronger theoretical justifications for the proposed relationships.

However, before the manuscript can be considered for publication, I recommend that the authors conduct a thorough language and grammatical review. Several sections still contain minor linguistic inconsistencies and awkward phrasings that may affect the overall readability and professionalism of the paper. A careful language edit, preferably by a native English speaker or professional editing service, is advised to enhance clarity and flow.

Once these minor language issues are addressed, I believe the paper will be suitable for publication.

7. PLOS authors have the option to publish the peer review history of their article (what does this mean? ). If published, this will include your full peer review and any attached files.

**Do you want your identity to be public for this peer review?** For information about this choice, including consent withdrawal, please see our Privacy Policy .

Reviewer #1: No

---

## [Author Response · Author response to Decision Letter 2]

24 Aug 2025

Dear Reviewers,

We sincerely thank you and the reviewer for taking the time to review our manuscript and for providing constructive comments. We appreciate your positive evaluation of the scientific quality of our work. We have carefully revised the manuscript according to your suggestions, and we believe these revisions have improved its clarity and readability. Below, we provide a detailed response to each comment.

Academic Editor’s Comments

Comment:

Thank you for your revision; a minor revision is requested to improve the overall English quality, particularly focusing on fluency, clarity, grammar, and consistent, precise word choice.

Response:

We have carefully proofread the entire manuscript for grammar, clarity, and consistent word choice. All linguistic inconsistencies and awkward phrasings have been corrected to improve the fluency and readability of the text. A professional language editing service was also used to ensure the manuscript meets publication standards.

Reviewer #1 Comments

Comment 1:

The authors have adequately addressed the previous comments and made substantial improvements to the theoretical framework, hypotheses development, and conceptual clarity. The revised manuscript reflects a clearer distinction between constructs and presents stronger theoretical justifications for the proposed relationships.

However, before the manuscript can be considered for publication, I recommend that the authors conduct a thorough language and grammatical review. Several sections still contain minor linguistic inconsistencies and awkward phrasings that may affect the overall readability and professionalism of the paper.

Response:

We sincerely thank the reviewer for the positive evaluation of our revisions. In accordance with your suggestion, we have conducted a thorough language and grammatical review of the manuscript. All sections have been carefully revised to eliminate inconsistencies and awkward phrasing, ensuring improved readability, clarity, and professionalism.

Comment 2:

Once these minor language issues are addressed, I believe the paper will be suitable for publication.

Response:

We greatly appreciate the reviewer’s acknowledgment. The manuscript has been fully revised for language quality, and we believe it is now ready for publication.

Reference List

If any references were updated, replaced, or corrected, include a brief statement here:

Response:

All references have been reviewed and updated as necessary to ensure accuracy and compliance with PLOS ONE guidelines. No retracted articles are cited; all references are current and relevant.

We thank the editor and reviewer once again for their valuable comments and guidance. We hope that the revisions we have made address all concerns and that the manuscript is now suitable for publication in PLOS ONE.

Sincerely,

Sun

---

## [Editor Report · Decision Letter 2]

28 Aug 2025

The Influence of Investor Sentiment on the Chinese Stock Market amid COVID-19: An Event Study Analysis

PONE-D-24-37585R2

Dear Dr. Sun,

We’re pleased to inform you that your manuscript has been judged scientifically suitable for publication and will be formally accepted for publication once it meets all outstanding technical requirements.

Kind regards,

Jae Wook Song

Academic Editor

PLOS ONE
---

## [Editor Report · Acceptance letter]

PONE-D-24-37585R2

PLOS ONE

Dear Dr. Sun,

I'm pleased to inform you that your manuscript has been deemed suitable for publication in PLOS ONE. Congratulations! Your manuscript is now being handed over to our production team.

Kind regards,

on behalf of

Professor Jae Wook Song

Academic Editor

PLOS ONE